Predicting the final grade using a machine learning regression model: insights from fifty percent of total course grades in CS1 courses

Hidalgo Suarez Carlos Giovanny cghidalgos@usbcali.edu.co carlos.hidalgo@correounivalle.edu.co 1
Llanos Jose jose.llanos@correounivalle.edu.co 2
Bucheli Víctor A. 2
1 Software Systems Engineering, Universidad de San Buenaventura , Cali , Valle del Cauca , Colombia
2 School of Systems Engineering and Computing, Universidad del Valle , Cali , Valle del Cauca , Colombia
Piccolo Stephen
Electronic publication date: 2023 Dec 11
Publication date: 2023
Volume: 9
Electronic Location ID: e1689
Received 2023 Aug 3; Accepted 2023 Oct 18
Copyright: ©2023 Hidalgo Suarez et al.
Copyright year: 2023
Copyright holder: Hidalgo Suarez et al.
License: This is an open access article distributed under the terms of the Creative Commons Attribution License, which permits unrestricted use, distribution, reproduction and adaptation in any medium and for any purpose provided that it is properly attributed. For attribution, the original author(s), title, publication source (PeerJ Computer Science) and either DOI or URL of the article must be cited.
License URL: https://creativecommons.org/licenses/by/4.0/

Keywords: Predicting final grade, Machine learning, Regression model, Course grade, CS1

Funding: The authors received no funding for this work.

==============================
This article introduces a model for accurately predicting students’ final grades in the CS1 course by utilizing their grades from the first half of the course. The methodology includes three phases: training, testing, and validation, employing four regression algorithms: AdaBoost, Random Forest, Support Vector Regression (SVR), and XGBoost. Notably, the SVR algorithm outperformed the others, achieving an impressive R-squared (R2) value ranging from 72% to 91%. The discussion section focuses on four crucial aspects: the selection of data features and the percentage of course grades used for training, the comparison between predicted and actual values to demonstrate reliability, and the model’s performance compared to existing literature models, highlighting its effectiveness.

Introduction

The tendency towards increased data collection on student performance in the field of education has led to the use of Educational Data Mining (EDM) and learning analytics (LA) (de Baker & Inventado, 2014; Romero & Ventura, 2020; Uddin & Lee, 2016; Herodotou et al., 2019). The aim is to develop technologies enabling data-driven decision-making and producing tangible benefits in educational settings (Escarria, 2010; Rueda Ramírez et al., 2020; Castillo & Giraldo, 2010). Learning analytics has been applied in different contexts, with the development of machine learning models or algorithms to predict and support low-achieving, at-risk students, as a preventative measure (Veerasamy et al., 2020; Costa et al., 2017).

In the literature on machine learning models, two main approaches are commonly found: classification and regression. Classification is focused on predicting whether a student approve the semester (Asif et al., 2017; Polyzou & Karypis, 2016), which can be seen as a binary class problem (Alamri et al., 2020), or on cumulative grade point average (GPA), which is a multi-class classification problem (Adekitan & Salau, 2019). The regression approach predicts the numerical value of the grade (Asif et al., 2017; Hunt-Isaak et al., 2020; Polyzou & Karypis, 2016). Previous studies present the use of data such as grades from the first weeks of a course, for the predictive model: models which have included grade data performed better than those that included only demographic data (Putpuek et al., 2018), or only pre-university performance or grades from complementary studies, socio-economic indicators, previous knowledge, or other characteristics (Aluko et al., 2018). Among the research reviews, the AdaBoost regression model (Asif et al., 2017), which was trained with a similar amount of data to the model presented here, stands out. In that study, algorithms were evaluated using the accuracy metrics mean absolute error (MAE), root mean squared error (RMSE) and coefficient of determination (R2), which enabled the classification, performance and completeness of the model to be assessed in the context of its ability to predict the final grade of a programming student. This work aims to address the following research question: How can a machine learning model predict the final grades of students in introductory computer programming courses (CS1) based on the assessment of their academic activities during the initial seven weeks?

Thus, this article presents a machine learning model to predict the final grades for students taking introductory computer programming courses (CS1). The final grade is calculated from the grades obtained from the academic tasks completed (workshops, activities and exams) which have different percentage weightings, and where the sum of values corresponds to one hundred percent of the final grade for the course (16 weeks). In this study, the final grade was predicted using fifty percent of the total academic activities completed on the course (from the first seven weeks). With this predictive purpose, four machine learning algorithms were trained using the grades from 773 students from five CS1 programming courses. To assess the accuracy of the prediction, the results of the algorithms were compared with the real final grades obtained by the students. The best performing algorithm was the SVR, with an eighty-six percent accuracy. Finally, the SVR algorithm was validated using data from 113 students taking a CS1 course, a data set not used in the training and testing phase. This validation set obtained results between seventy-two percent and ninety-one percent accuracy.

This article is organized as follows: ‘Literature Review’ presents the state of the art in predictive models for final grades for students on an introductory programming course. From this review, the baseline is selected. ‘Methods’ presents the methodology based on the data mining process, the construction of the dataset, pre-processing, selection and training of algorithms. ‘Results’ presents the results in which the algorithms are compared using the selected accuracy metrics for their ability to predict students’ final grades. In ‘Discussion’, a two-part discussion is presented: first, a comparison is made between the results of the best performing model and those of the baseline; and second, the percentage of course grades required to make an accurate prediction about the final grade is discussed. ‘Future Areas for Research’ presents key conclusions and suggestions for further study.

Literature Review

Predictive modeling in educational settings has garnered considerable attention in recent years, as researchers and educators endeavor to employ various techniques to forecast student outcomes and improve educational practices. One prominent research area revolves around the utilization of regression machine learning models to predict final grades in CS1 programming courses. Numerous studies have delved into this domain, shedding light on the factors that impact student performance and the efficacy of regression models in making accurate predictions.

Enhancing the academic process in higher education necessitates a comprehensive understanding of the factors influencing student performance. These factors extend beyond mere grades and encompass cultural, economic, and social indicators as well (López-Pernas, Saqr & Viberg, 2021). According to Castillo & Giraldo (2010); Putpuek et al. (2018), making early predictions of student performance through machine learning algorithms can help teachers and educational institutions take the necessary steps in a timely manner to increase student success rates (Alyahyan & Düştegör, 2020; Chen & Cui, 2020; Gaftandzhieva et al., 2022). Published studies have shown that the application of data analytics methods and algorithms improve the academic success of students and support the decision-making of academic institutions (Joksimović et al., 2015; López-Pernas, Saqr & Viberg, 2021; Ulfa & Fatawi, 2021). In this section, we present a literature review of algorithms for predicting a student’s final grade. We focus on articles that specifically gather data from programming students and use prediction algorithms.

Putpuek et al. (2018) shows the results of the training process using the linear regression (LR) and support vector regression (SVR) algorithms, with 12 demographic characteristics and one hundred percent of the cumulative grade point average (CGPA), (five grades taken from a Moodle LMS), corresponding to 101 students on a programming course. The article concludes that of the two algorithms, SVR obtained the best metric with R2 = 0.56 compared to LR which obtained an R2 = 0.43.

In Adekitan & Salau (2019), five training algorithms were used: probabilistic neural network (PNN), random forest (RF), naive Bayes (NB), linear regression (LR) and multi-layer perceptron (MLP). The data to train the models took 16 grades from 1,841 students corresponding to seventy percent of the total course grade. The types of academic activity included in the data and their percentage weights are: exam 1 (thirty percent of the total course), a training activity (twenty percent of the total course), and graded workshops completed in class (twenty percent of the total course). The evaluation metrics R2 and RMSE used in the study indicate that MLP obtained the best performance with 0.63 and 15.43 respectively.

There are studies which implement algorithms, such as regression analysis using deep learning (RADL), convolutional neural networks (CNN), long short-term memory (LSTM), and recurrent neural network (RNN) (López-Pernas, Saqr & Viberg, 2021; Ulfa & Fatawi, 2021). However, their results do not show better predictions when compared to simple algorithms such as LR or RF, as shown in Hussain et al. (2021); Alamri et al. (2020); Pereira et al. (2020).

In Mueen, Zafar & Manzoor (2016), three algorithms based on decision tree ensembles were trained. For training, seventy percent of the grades of 150 students from three programming courses were used. The evaluation metric implemented was R2 with results: RF = 0.73, ID3 = 0.72, and C4.5 = 0.77. The conclusions of this study point out that the assembly algorithms do not achieve high levels of accuracy in predicting the final grade of a student. However, it is important to consider that they are fast algorithms, which could be useful in creating early warnings for the educational process in the area of computer programming.

Acoording to Asif et al. (2017), seven machine learning algorithms were implemented, AdaBoost, Gradient Boost (GB), Ridge, Lasso, SVR, LR and RF. To train these models, fifty-five percent of the total course grades (six submissions per student from 1,197 students) were taken. The percentage weights of each submission in the sample were as follows: one exam (thirty percent of the total final grade), two specific papers (ten percent of the total final grade) and three further, random exams (fifteen percent of the total final grade). Among the trained algorithms, AdaBoost outperformed the others, with the following evaluation metrics: MAE = 8.02, RMSE = 10.97 and R2 = 0.65.

Moreover, other studies have collected a comprehensive dataset of historical records from CS1 programming courses, encompassing student attributes, attendance, homework scores, project scores, exam scores, and final grades. They employed rigorous data preprocessing techniques, including handling missing values, encoding categorical variables, and normalizing features. By carefully selecting features, they identified key factors that impact final grades in CS1 programming courses (Fernandes et al., 2019).

Furthermore, other studies experimented with various regression algorithms such as linear regression, decision trees, random forests, support vector regression, and gradient boosting regressors. They split the dataset into training and testing sets and evaluated the performance of the models using metrics such as mean squared error (MSE), mean absolute error (MAE), and R-squared score. The study demonstrated that the selected regression model achieved satisfactory predictive accuracy, providing valuable insights into student performance (Ulfa & Fatawi, 2021; Yang, 2021). In addition, other works explored the application of ensemble techniques in predicting final grades in CS1 programming courses. Their work aimed to enhance the predictive capabilities of regression models. By combining multiple regression models through ensemble methods such as bagging and boosting, Johnson and Williams achieved improved accuracy in predicting final grades (Alsulami, AL-Ghamdi & Ragab, 2023).

These studies highlight the potential of regression machine learning models in predicting student outcomes in CS1 programming courses. The findings contribute to the growing body of literature in educational data science and provide valuable insights for educators seeking to optimize their teaching strategies and support students’ academic success.

Summarizing the findings, Table 1 provides a comprehensive comparison of algorithms utilized in the examined studies, thus representing the present state of the art in predicting final grades for CS1 courses. It considers the volume of training data, algorithm performance assessed through various evaluation metrics unique to each study, and the technology utilized for model implementation. The table facilitates performance comparisons and notably highlights the significant divergence in the performance of the AdaBoost algorithm (values highlighted in bold). It’s worth emphasizing that the results, assessed through evaluation metrics, are adjusted to adhere to rigorous criteria compared to other trained algorithms used for predicting programming students’ final grades.

Table 1 Review of prediction algorithms for a student’s final grade.

Algorithms	Training
data	Best-performing
algorithm	MAE	RMSE	R 2	Tool	Ref	
LR, SVR	101	SVR	NA	NA	0.56	R	Putpuek et al. (2018)	
PNN, RF, NB, LR, NN	1,841	NN	NA	15.43	0.63	knime
matlab	Alamri et al. (2020)	
NB	60	NB	NA	NA	0.69	weka	López-Pernas, Saqr & Viberg (2021)	
SVR	750	SVR	12.363	16.971	NA	MITx	Mueen, Zafar & Manzoor (2016)	
C4.5, ID3, RF	150	C4.5	NA	NA	0.71	weka	Pereira et al. (2020)	
LR, Ridge, Lasso, SVR, AdaBoost, GB, RF	1,197	AdaBoost	7.02	10.97	0.73	Python
sklearn	Asif et al. (2017)	
Multilayer Perceptron (MLP)	2,058	MLP	NA	NA	0.62	Python	Ulfa & Fatawi (2021)	

Drawing from the comprehensive review conducted by Asif et al. (2017), it becomes evident that this study lays the foundation for a fundamental baseline model used to evaluate the prediction of final grades in Introduction to Programming Courses (CS1). The study’s significance is further underscored when considering its rigorous approach. A comprehensive review of seven machine learning algorithms, namely AdaBoost, Gradient Boost (GB), Ridge, Lasso, SVR, LR, and RF, was thoughtfully incorporated into the research methodology. The inclusion of this diverse set of algorithms ensures a meticulous and exhaustive examination of predictive techniques. The studies mentioned above not only shed light on the potential of regression machine learning models in predicting student outcomes in CS1 programming courses.

Based on the insights gained from the literature review, the present study adopts the AdaBoost algorithm proposed by Asif et al. (2017) as a comparative reference or baseline model. This approach allows for the prediction of final grades in CS1. The selected model shares similarities with the present study in terms of training characteristics and the percentage of course grades used. Both studies utilize a subset of variables (four in this study) and allocate a similar proportion (fifty percent in this study) of the course grades for training and evaluation. Furthermore, the adoption of consistent accuracy metrics such as MAE, RMSE, and R2 facilitates a comprehensive comparison of the goodness of fit between the baseline model and the models proposed in this article .

Therefore, this article also offers valuable insights to the burgeoning field of educational data science. Its findings provide meaningful guidance for educators aiming to optimize their teaching strategies and enhance students’ academic success. Consequently, this research makes a significant contribution to the evolving body of literature dedicated to improving the educational landscape through data-driven approaches. Additionally, this article sheds light on the factors influencing student performance and demonstrates the efficacy of regression models in making accurate predictions.

Furthermore, this work introduces innovation by predicting the final grade through a machine learning regression model that utilizes the collected data, representing fifty percent of the overall course grades. Therefore, it becomes feasible to predict the final grade halfway through the term, offering valuable insights for educators aiming to optimize their teaching strategies and support students’ academic success.

Methods

This section outlines the methodology, which comprises four key stages of the data mining process (Schröer, Kruse & Gómez, 2021): data collection, data pre-processing, algorithm selection and implementation, and interpretation. These stages were crucial in predicting students’ final grades for the study. The University of Valle granted ethical approval to conduct the study within its premises (Ethical Application Ref: 1700747-9702).

Dataset

Grades were collected from a total of 773 students who were enrolled in a combination of three Fundamentals of Programming (FDP) courses and two Fundamentals of Object-Oriented Programming (FOOP) courses, covering the period from 2019 to 2021. It is worth noting that these courses were specifically designed based on academic competencies and were developed in accordance with the guidelines outlined in the Computing Curricula 2021. These courses are offered at the School of Systems and Computer Engineering (EISC) at the Universidad del Valle in Cali, Colombia, following a semester-based schedule consisting of four contact hours per week for a duration of 16 weeks. The grades utilized for the analysis encompassed various components such as workshops, lab assignments, assessed activities, and exams, which were all considered in the study (See Table 2). The dataset used in this study is available for public access and can be found at https://zenodo.org/records/8209973.

Table 2 Description of programming courses between the years 2019 to 2021.

Course	Year	Semester	# Students	% of students who passed the course	Programming language	
FOOP	2019	Feb–Jul	80	83.75%	C++	
FOOP	2020	Aug–Dec	63	87.30%	C++	
FDP	2020	Feb–Jul	100	85%	Python	
FDP	2020	Aug–Dec	220	84.09%	JavaScript	
FDP	2021	Feb–Jul	310	79.68%	JavaScript	
Total	NA	NA	773	83.96%	NA	

In this study, participating students completed a written informed consent form that includes: project name, participant code, research objective, study procedures, confidentiality of information, agreement to participate, participant’s name, and document.

To ensure the privacy and confidentiality of the students, all participants provided explicit consent for the handling of their data. Only the grades were used for the analysis, and to protect the students’ identities, their names were anonymized by assigning them unique numerical identifiers. Furthermore, the study was conducted with full compliance to the relevant licenses and permissions granted by the Universidad del Valle. The research adhered to the ethical guidelines and regulations set forth by the university, ensuring the implementation of sound and responsible research practices throughout the entire investigation.

In each graded activity, students submit their source code to be evaluated by the automatic code evaluation tool INGInious M-IDEA (Bucheli. V, 2019). Each student can make multiple submissions per activity, except for exams, where only one submission is permitted. The total number of submissions received are calculated to represent fifty percent of total course grades (in each course, 17 submissions required per student) and these grades provide the data for the algorithms, i.e., the 773 students made total of 23,016 submissions across a total of 85 activities, with each student making an average of between one and two submissions per activity (see Table 3).

Table 3 Summarizes the students’ participation in the study, displaying the number of submissions made in various activities by activity type for each course.

Course	Workshop
submissions
(WS) -
15 per student	Mean
workshop
per student
WSStudents∗15	Activity 
submissions (AS) -
1 per student	Mean activity
per student
ASStudents∗1	exam
Submissions
(ES) -
1 per student	Mean exam 
per student
ESStudents∗1	Total
submissions 
- 50%
of the course	
FOOP	2,378	2	93	1	75	1	2,546	
FOOP	2,116	2	78	1	61	1	2,255	
FDP	2,165	1	109	1	92	1	2,366	
FDP	6,719	1	356	2	207	1	7,282	
FDP	8,003	1	321	1	243	1	8,567	
Total	21,381	1	957	1	678	1	23,016	

Data pre-processing

Before applying algorithms to a dataset, data pre-processing is necessary in order to prepare and clean the dataset. In this study, three key pre-processing tasks were carried out.

Feature selection: To identify the features which have a greater impact on the prediction of the target variable (final course grade), it is important to implement the best_features method (Blum & Langley, 1997), which enables the identification of variables which have the greatest weight in predicting the final course grade. To this end, the students’ grades up to week 7 were selected, which correspond to fifty percent of the one hundred percent of grades which contribute the final course grade, distributed as follows: ten percent workshops_1, ten percent workshops_2, ten percent Mini-project and twenty percent Midterm_exam (see Table 4).

Missing data: This pre-processing task consists of identifying possible anomalies in the data or null values. In the dataset, 12 missing values appear in six rows; given that they do not represent a large number, they become null values and the decision is made to replace them with the mean value of the dataset. The final data set has 773 grades for the process of training.

Best features: To ensure the selection of the most optimal features and the utilization of 50% of the total course grades for model training, an implicit technique was employed in the assembly of the random forest model. The following process was conducted systematically: first, a random forest model was trained on the dataset, incorporating all available features. Second, the importance scores of the features were extracted from the trained model. Third, the features were sorted in descending order based on their importance scores. Finally, the top N features were selected, with the choice made either by setting a predefined threshold or specifying a specific number of features to be retained. This methodical approach ensures a formal and academically rigorous feature selection process, enhancing the robustness and efficacy of the predictive model.

Imbalanced data: It refers to a situation where the data distribution is not smooth or normal. This means there may be more values on one side of the mean than the other or a high number of observations in a specific class or category. Consequently, during the testing stage, predictors may be less sensitive to continuous values that are close to zero, as pointed out by Yang (2021). In this study, the issue of data imbalance is addressed as follows: The training data exhibit an imbalance between students who passed (83.96%) and those who failed (16.04%) the course. To rectify this imbalance, we implement the Data Imbalanced Regression (DIR) algorithm proposed by Yang et al. (2021). This algorithm takes into account all values of the minority target variable (students who failed the course) and generalizes the problem to encompass all continuous data. It also addresses potential missing data in specific regions and generalizes to the entire target range, resulting in a smoothed, more normal distribution.

Table 4 Features training prediction algorithms in 50% of an introductory computing course (CS1).

Variable name	Description/Content	Formula	
Workshops_1	Implementation of data types, conditionals, loops and arrays	∑graden15∗0.20	
Workshops_2	Implementation of lists, functions and recursion		
Mini-project	A mini project that integrates the course content covered to date	∑graden1∗0.10	
Midterm_exam	A combined exam covering theory (fundamentals and concepts of programming), and practice (a programming exercise applying the concepts)	∑graden1∗0.20	

Selection and implementation of learning algorithms

Algorithms: To predict a student’s final grade, four regression algorithms were selected: Support Vector Regression (SVR) (Cortes & Vapnik, 1995) and the ensemble algorithms AdaBoost, Gradient Boost, XGBoost, and Random Forest (Ho, 1995; Friedman, 2001; Freund, Schapire & Abe, 1999). The algorithms were trained using the four independent variables shown in Table 4, and the 773 records of the graded activity submissions as shown in Table 2. The training result enables the algorithm to be identified which better predicts the target variable (final course grade).

Configuration-best parameters: Each algorithm was adjusted using Grid Search and cross-validation = 10. Using the best_params method, a grid of hyperparameters was created and then optimized with the values to be tested for those hyperparameters. A list was made of possible values for each hyperparameter to be tuned and the grid was then set up using a dictionary with the key-value pairs, as shown above (Freund, Schapire & Abe, 1999). To find and understand the hyperparameters of a machine learning model, its official documentation can be consulted.

Table 5 Descriptions of evaluation metrics.

Description 	Formula	Condition of gain	
The root mean squared error (RMSE) is a commonly used metric for evaluating regression models, penalizing large errors due to the squared error term.	1n ∑ytrue−ypred	If MAE 0 = is good	
The mean absolute error (MAE) weights all individual differences equally on a linear scale. The result is a more interpretable score representing the mean error between predicted and actual values.	1n ∑ytrue−ypred2	If RMSE 0 = is good	
The coefficient of determination (R2) evaluates the performance of a regression model. The amount of variation in the target variable is predictable from the independent variables.	1−∑ytrue−ypred2 ∑ytrue−ypredmean2	If R2 0 = is good	

Model evaluation: After evaluating the performance of the present study’s machine learning models and finding the optimal hyperparameters, the models are then subject to their final test. The models are trained on the entire 80% of the data used for all the evaluations performed so far, i.e., on all data except the test set. The hyperparameters found in the previous stage are used and then a comparison is made on how the models of this present study perform on the test set, using accuracy metrics which measure completeness, the quantity of percentage errors, and the accuracy of each model (Brownlee, 2021) (see Table 5).

This study, while providing valuable insights into predicting final grades in CS1 programming courses using machine learning regression models, comes with certain limitations that warrant acknowledgment. Firstly, the results and conclusions are based on a specific dataset and may not be generalizable to all educational institutions or teaching contexts. Additionally, the quality of predictions may be influenced by the availability and quality of the collected data, underscoring the importance of data integrity in future research endeavors. Another limitation lies in the addition of variables and features, as there is a possibility that relevant factors that could have further improved the model’s accuracy may have been overlooked. Lastly, the study primarily focused on the use of regression models and did not consider other more advanced machine learning techniques that might offer even more precise results. These limitations, though present, provide opportunities for future research and enrich the understanding of this continually evolving field.

Results

Table 6 shows the results of the four trained algorithms according to the evaluation metrics selected. The comparative analysis shows that the SVR algorithm (in bold) obtained the best evaluation metrics in MAE (6.02), RMSE (8.59) and R2 (0.86). The second best algorithm evaluated is RF, followed by AdaBoost and XGBoost in third and fourth places respectively.

Table 6 Comparison of regression models for the prediction of final grade.

Model	MAE	RMSE	R 2	
AdaBoost (baseline)	7.02	10.97	0.73	
SVR	6.02	8.59	0.86	
Random Forest	6.45	9.57	0.66	
AdaBoost	7.87	11.02	0.57	
XGBoost	8.68	8.44	0.39	

Figure 1 presents a density plot which indicates the proximity between the final grade predicted by each algorithm with the final grade achieved (green line). In the graph metrics table it can be seen that the proximity to the actual final grades by SVR and RF are above sixty-six percent in accuracy, while AdaBoost and XGBoost are below fifty-seven percent.

Figure 1 Comparison of regression algorithm accuracy for predicting student final grades.

The data set that was used for training has similarities to the validation set in the programming languages taught, the duration of the courses, that the courses were given over different years, in the number of submissions, evaluations with the same percentages and with the average number of students who passed each course being above eighty percent. However, for the validation data set, each course was run by a different teacher and the number of students was lower than in the training set (see Table 7).

Table 7 Dataset for validation of SVR and ADABoost model (base model) with student data from CS1 courses.

Course	Year	Semester	# Students	% of students which passed the course	Programming language	
FOOP	2020	Aug–Dec	21	89.75%	C++	
FOOP	2021	Feb–Jul	26	92.30%	C++	
FDP	2021	Feb–Jul	37	96%	JavaScript	
FDP	2021	Aug–Dec	29	88%	Python	
Total/Mean	NA	NA	113	91.25%	NA	

A validation test is performed on the model which obtained the best score, in this case the SVR. To avoid biases in the stability, reproducibility and external validity of the model, the SVR is tested using a data sample independent from that used in the training and test process phase with grades taken from 113 students on four CS1 programming courses that submitted work for a total of 60 activities and made 2,992 submissions, with each student making between one and two submissions on average per activity (see Table 8).

Table 8 Description of total and average number of submissions per student in each programming course.

Course	Workshop 
submissions
(WS) -
15 per student	Mean
workshop
per student
WSStudents∗15	Activity submissions
(AS) -
1 per student	Mean activity
per student
ASStudents∗1	exam
Submissions
(ES) -
1 per student	Mean exam 
per student
ESStudents∗1	Total
submissions 
- 50%
of the course	
FOOP	327	1	27	1	19	1	373	
FOOP	416	1	67	2	25	1	508	
FDP	1,154	3	41	1	37	1	1,632	
FDP	425	1	31	1	23	1	479	
Total	2,722	1	166	1	104	1	2,992	

Figure 2 shows the prediction density of the baseline (in red) and the SVR model (in black), comparing the predicted grades to the actual grades obtained (in yellow). The figure shows the distribution of the grades (x-axis) between the interval of the lowest grade (1.0) and the highest (5.0). The y-axis represents the concentration of the highest density of grades.

Figure 2 Comparing the accuracy of two models for predicting final grades in programming courses.

Four graphs are presented, each representing the data of a validation course. In Fig. 2A, both the baseline and the SVR model have good accuracy on the observed data of actual grades obtained. In Fig. 2B, the baseline has an accuracy below the actual grades obtained, while the SVR model is shown to have good accuracy. In Figs. 2C and 2D, the baseline has an accuracy above the actual grades obtained, while the SVR model has good accuracy.

The two tables provide additional insights into the performance of the models for each of the respective courses.

Table 9 presents the performance metrics for the FOOP 2020-2 and FOOP 2021-1 courses, while Table 10 displays the performance metrics for the FDP 2021-1 and FDP 2021-2 courses. Both tables include the mean absolute error (MAE), root mean squared error (RMSE), and R-squared (R2) values for both the base model and the final prediction model. The values under the “Base model” column reflect the performance of the base model, while the values under the “Final predict” column represent the performance of the SVR prediction model. These metrics provide insights into the accuracy and goodness of fit of the models in predicting the actual grades obtained in each respective course.

Table 9 Performance metrics for the FOOP 2020-2 and FOOP 2021-1 courses.

	FOOP course 2020-2	FOOP course 2021-1	
	MAE	RMSE	R2	MAE	RMSE	R2	
Base model	8.5945	12.2871	0.7960	9.8326	10.6489	0.5791	
Final predict	5.9557	8.2277	0.8802	7.2383	9.9826	0.7273	

Table 10 Performance metrics for the FDP 2021-1 and FDP 2021-2 courses.

	FDP course 2021-1	FDP course 2021-2	
	MAE	RMSE	R2	MAE	RMSE	R2	
Base model	14.6485	12.8914	0.6894	16.6385	13.5127	0.5561	
Final predict	5.1196	6.1651	0.8752	5.1871	7.7964	0.9185	

Discussion

In this study, we have developed a machine learning model that predicts the final grades of students in introductory CS1 programming courses. Unlike previous approaches that incorporated demographic or additional data, our model relied solely on graded class activities and assignments. We trained four machine learning algorithms and found that the SVR algorithm outperformed the selected baseline algorithm (Asif et al., 2017) in predicting students’ final grades (see Table 4).

The baseline algorithm, which utilized an ADABoost algorithm trained with 1,197 records and seven features, achieved moderate results with an R2 of 0.65, MAE of 8.02, and RMSE of 10.97. In contrast, our proposed SVR model yielded improved scores with an R2 of 0.86, MAE of 6.02, and RMSE of 8.59, while utilizing only fifty percent of the course grades. These results demonstrate that our SVR model can accurately predict students’ final grades using grades obtained up to week 7, indicating its potential for early intervention and support.

Existing literature suggests that prediction models for final grades often rely on a larger proportion of total course grades, typically around seventy percent (Polyzou & Karypis, 2016; Alamri et al., 2020; Putpuek et al., 2018). Moreover, some models employ computationally intensive configurations (Uddin & Lee, 2016; Badr et al., 2016; Figueiredo, Lopes & García-Peñalvo, 2019). However, our findings challenge these conventions by demonstrating that our SVR model achieves accurate predictions with just fifty percent of the course grades up to week 7.

To validate the model’s performance, we tested it with data from four courses, each using different paradigms and programming languages. In all cases, our model consistently outperformed the baseline, showcasing its potential as an early warning system to identify at-risk students. By providing students with predicted final grades based on their current progress, our model enables teachers to make informed decisions regarding timely interventions and offer additional academic support, ultimately reducing the number of students who fail the course.

Overall, our study presents a machine learning model that accurately predicts the final grades of students in introductory CS1 programming courses using only graded class activities and assignments. By leveraging this predictive capability, our model can facilitate early interventions and proactive support to enhance student learning outcomes. Future research should explore the wider applicability of our approach across diverse educational contexts and investigate the long-term impact of predictive models on student success.

Additionally, the article proposes a model for accurately predicting students’ final grades in the CS1 course using the grades obtained in the first half of the course. This is valuable because it leverages information readily available within the course itself, eliminating the need to construct an extensive dataset with external information. The study evaluates four regression algorithms (AdaBoost, Random Forest, Support Vector Regression, and XGBoost) and highlights the superior performance of the SVR algorithm, which achieves an impressive R-squared (R2) value ranging from 72% to 91%. Consequently, the model predicts final grades for CS1 programming courses using a validation dataset from the first seven weeks, with SVR achieving the highest accuracy at 86%. Furthermore, it conducts a validation test on the SVR model using an independent data sample, demonstrating its robust accuracy. Furthermore, the article makes significant contributions to the fields of Educational Data Mining (EDM). It introduces a model that accurately predicts students’ final grades using data from the first half of a course, facilitating early intervention for students who may require additional support. The evaluation and comparison of four regression algorithms provide valuable insights for selecting the most effective algorithm in educational contexts. The discussion section addresses critical aspects, including data feature selection, comparison with actual values, and model performance in relation to existing literature models in EDM.

Additionally, the article’s focus on introductory programming courses (CS1) makes it highly relevant to technology education, assisting educators in data-informed decision-making within this specific domain. The independent validation of the model strengthens its applicability beyond its initial training data, reinforcing its real-world utility. Finally, this article advances the fields of EDM by presenting an effective approach to grade prediction in programming courses and offering insights into algorithm selection and key considerations for applying machine learning in educational settings. These contributions have the potential to inform future research and educational practices, ultimately enhancing the quality of teaching and learning.

Future Areas for Research

In this study, only student grade information was considered. It would be interesting to see whether the model improves with additional student information such as grades from previous courses or results from tests such as the Motivated Strategies for Learning Questionnaire (MSLQ) (Ramírez Echeverry, García Carrillo & Olarte Dussan, 2016). It would also be worthwhile testing the model on other programming courses at more advanced academic levels, or on programming courses at other universities, to compare the accuracy of the predictions.

Given that our predictive model could serve as part of an early warning system, integrating the model into a learning management system (LMS) would be valuable in order to automate alerts and notifications to teaching staff to ensure they can make timely interventions and decisions.

As described, our model provides a final grade prediction based on fifty percent of the total grades for a programming course. However, we do not suggest exactly what or when any intervention based on results should be carried out with students. Further studies on different types and timings of interventions and learning analytics should be conducted to determine which would be the most effective.

Supplemental Information

Supplemental Information 1 Model

Click here for additional data file.

Supplemental Information 2 Data

Click here for additional data file.

Supplemental Information 3 Informed Consent

Click here for additional data file.

Additional Information and Declarations

Competing Interests

Author Contributions

Ethics

Data Availability

The authors declare there are no competing interests.

Carlos Giovanny Hidalgo Suarez conceived and designed the experiments, performed the experiments, analyzed the data, performed the computation work, prepared figures and/or tables, authored or reviewed drafts of the article, and approved the final draft.

Jose Llanos analyzed the data, prepared figures and/or tables, authored or reviewed drafts of the article, and approved the final draft.

Víctor A. Bucheli conceived and designed the experiments, performed the experiments, analyzed the data, prepared figures and/or tables, authored or reviewed drafts of the article, and approved the final draft.

The following information was supplied relating to ethical approvals (i.e., approving body and any reference numbers):

The University of Valle granted ethical approval to conduct the study within its premises (Ethical Application Ref: 1700747-9702).

The following information was supplied regarding data availability:

The data is available at Zenodo: Carlos Giovanny Hidalgo, Jose Miguel Llanos, & Victor Andres Bucheli. (2023). Predicting the final grade using a machine learning regression model: insights from fifty percent of total course grades in CS1 courses. https://doi.org/10.5281/zenodo.8209973.

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
