# Peer review of "Predicting the final grade using a machine learning regression model: insights from fifty percent of total course grades in CS1 courses"

_PeerJ Computer Science, doi:10.7717/peerj-cs.1689_

## Round 0.1 · original submission · Minor Revisions

Two of the reviewers suggested minor revisions. I wanted to get one additional person to look at it, so there are three reviewers now. Because of this, there are some additional suggestions to improve the manuscript. Please address those as well as you can and/or provide strong justification on why they are not important.

·

Basic reporting

The study is an interesting one. I have only a few suggestions.
1. Major contribution of the study should be highlighted.
2. Research gaps should be elaborated more.
3. Novelty of the study should be explained.

Experimental design

1. What were the strategies applied by the authors to tackle the data imbalance issue?

2. Comparison of the results with the SOTA studies in the domain may be incorporated.

Validity of the findings

1. The limitation of the study may be discussed.

Additional comments

The authors should cite some of the recent studies in the domain. Some of the studies listed below. The authors may not necessarily required to cite them.

1. Gaftandzhieva, S., Talukder, A., Gohain, N., Hussain, S., Theodorou, P., Salal, Y. K., & Doneva, R. (2022). Exploring online activities to predict the final grade of student. Mathematics, 10(20), 3758.

2. Hussain, S., Gaftandzhieva, S., Maniruzzaman, M., Doneva, R., & Muhsin, Z. F. (2021). Regression analysis of student academic performance using deep learning. Education and Information Technologies, 26, 783-798.

·

Basic reporting

The manuscript improved a lot since the first review. Congratulations! However,
I still have a few remarks:

1. The reference style is inconsistent. Sometimes inline citations are formatted
entirely between brackets, sometimes only the year is in brackets. The
reference on line 119 is missing the year. Repeating the title of the work
and the location it was published is also not necessary (as done e.g. on line
112 or line 126).
2. The start of the sentence on line 135 is not grammatically correct.
3. It would be better to introduce the research question in the introduction.
4. The link on lines 185-186 is not well-formatted.

Experimental design

no comment

Validity of the findings

no comment

Reviewer 3 ·

Basic reporting

I consider that the observations have been results.

Experimental design

I consider that the observations have been results.

Validity of the findings

I consider that the observations have been results.

Additional comments

I don't have additional comments.

---

## Round 0.2 · accepted · Accept

The reviewers were happy with your changes. One reviewer identified some grammatical issues. Please look at those and address them in the final version that goes to production.

·

Basic reporting

All my concerns are addressed well.

Experimental design

It is already suitable.

Validity of the findings

It is also found to be ok.

·

Basic reporting

1) A number of grammatical changes look incorrect to me:
The change of "passes" to "approve" on line 33 is incorrect.
The change of "In" to "According to" on line 112 is incorrect.

Experimental design

no comment

Validity of the findings

no comment